# One-Step Co-Electrodeposition of Copper Nanoparticles-Chitosan Film-Carbon Nanoparticles-Multiwalled Carbon Nanotubes Composite for Electroanalysis of Indole-3-Acetic Acid and Salicylic Acid

**DOI:** 10.3390/s22124476

**Published:** 2022-06-13

**Authors:** Yiwen Kuang, Mengxue Li, Shiyu Hu, Lu Yang, Zhanning Liang, Jiaqi Wang, Hongmei Jiang, Xiaoyun Zhou, Zhaohong Su

**Affiliations:** 1College of Bioscience and Biotechnology, Hunan Agricultural University, Changsha 410128, China; kuangyiwen@stu.hunau.edu.cn; 2College of Chemistry and Materials Science, Hunan Agricultural University, Changsha 410128, China; limengxue@stu.hunau.edu.cn (M.L.); hushiyu@stu.hunau.edu.cn (S.H.); yanglu00@stu.hunau.edu.cn (L.Y.); liangzhanning@stu.hunau.edu.cn (Z.L.); wangjiaqi@stu.hunau.edu.cn (J.W.); jhmndcn@hunau.edu.cn (H.J.)

**Keywords:** carbon nanoparticles, copper nanoparticles, chitosan film, multiwalled carbon nanotubes, electroanalysis of indole-3-acetic acid and salicylic acid

## Abstract

A sensitive simultaneous electroanalysis of phytohormones indole-3-acetic acid (IAA) and salicylic acid (SA) based on a novel copper nanoparticles-chitosan film-carbon nanoparticles-multiwalled carbon nanotubes (CuNPs-CSF-CNPs-MWCNTs) composite was reported. CNPs were prepared by hydrothermal reaction of chitosan. Then the CuNPs-CSF-CNPs-MWCNTs composite was facilely prepared by one-step co-electrodeposition of CuNPs and CNPs fixed chitosan residues on modified electrode. Scanning electron microscope (SEM), transmission electron microscopy (TEM), selected area electron diffraction (SAED), energy dispersive spectroscopy (EDS), X-ray diffraction (XRD), Fourier transform infrared spectroscopy (FT-IR), cyclic voltammetry (CV), electrochemical impedance spectroscopy (EIS), and linear sweep voltammetry (LSV) were used to characterize the properties of the composite. Under optimal conditions, the composite modified electrode had a good linear relationship with IAA in the range of 0.01–50 μM, and a good linear relationship with SA in the range of 4–30 μM. The detection limits were 0.0086 μM and 0.7 μM (*S/N* = 3), respectively. In addition, the sensor could also be used for the simultaneous detection of IAA and SA in real leaf samples with satisfactory recovery.

## 1. Introduction

Phytohormones are small molecular substances synthesized by plants themselves. Indole acetic acid (IAA) and its derivatives are important plant growth hormones. They are involved in the regulation of various biological processes such as plant cell elongation and reproduction, leaf and flower withering, and plant vascular tissue decomposition. They have different important functions in various stages of plant growth and development. Salicylic acid (SA), another phytohormone, is involved in the regulation of many physiological processes, such as flowering, heat production, senescence, and autophagy, and also plays an important role in abiotic stresses such as low temperature, high temperature, and salt [1]. The content of IAA and SA in plants fluctuates, and the concentration of IAA is about 40–160 ng/g [2,3]. The concentration of SA is about 100~200 ng/g [4], which can complete normal growth and development. External factors (salt, water, temperature, etc.) can lead to abnormal levels of IAA and SA in plants, which can be regulated by applying exogenous phytohormones to improve plant growth. Therefore, it is necessary to establish a method for the determination of IAA and SA with high selectivity and sensitivity. The current detection methods include liquid chromatography-mass spectrometry [5,6], molecular imprinting method [7,8], capillary electrophoresis [9], fluorescence spectroscopy [10] and electrochemical method [11]. Among them, electrochemical method is favored for its advantages of simple operation, high sensitivity, and fast analysis compared with other methods. 

Chitosan (CS) [12] is a green material with biological activity, low toxicity, biocompatibility, and biodegradability. It is an amino polysaccharide derived from chitin. When pH < pKa (pKa = 6.3), most of the amino groups are protonated, making chitosan a water-soluble cationic polyelectrolyte. When pH > pKa, the amino group of chitosan is deprotonated and becomes insoluble in water. According to this characteristic, chitosan films can be formed by electrodeposition [13,14]. In recent years, CS derivatives, CS composite films [15], and CS-based nanoparticles [16] have also been widely studied. Among them, CS-based nanoparticles are the research focus in recent years. Compared with CS, CS-based nanoparticles have the advantages of volume effect, surface effect, quantum size effect, and dielectric confinement effect of nanomaterials, which have attracted wide attention due to their large specific surface area [17]. CS-based nanoparticles are widely used in food and agriculture [18]. At present, no one uses CS-based nanoparticles fixed chitosan residues to prepare polymer film by electrochemical method for electroanalysis application.

Multiwalled carbon nanotubes (MWCNTs) have been widely used in sensors due to their large surface area, good conductivity, and chemical stability. The solubility of MWCNTs in an aqueous solution is not good [19], considering the dispersion of MWCNTs, other materials are used to composite with them [20,21]. In recent years, electrochemical sensors based on carbon nanotube composites (carboxymethyl cellulose-montmorillonite-single-walled carbon nanotubes [1], MWCNTs-carbon black composites [22], MWCNTs-CS [23]) for detection of IAA and SA. Herein, we combine carbon nanoparticles with MWCNTs, which can not only improve the dispersion of materials, but also improve their electronic transmission capacity. In addition, compared with other metal materials, such as copper nanoparticles (CuNPs) [24] are cheap, which are conducive to large quantities of actual detection, and can make the material on the electrode surface not easy to fall off.

In this paper, using CS as the carbon source, carbon nanoparticles (CNPs) were prepared by a hydrothermal method. Then the CNPs were ultrasonically mixed with MWCNTs and drip-dry on the surface of a glassy carbon electrode (GCE) to obtain CNPs-MWCNTs/GCE. It was then placed in CuSO_4_ solution and electrodeposited at −0.4 V to obtain CuNPs. At the same time, due to the electrolysis of water at the same potential, there is a relatively high pH region near the electrode, so that CNPs fixed chitosan residues form a chitosan film (Figure 1) by electro-deprotonation. Finally, a new CuNPs-CSF-CNPs-MWCNTs composite was prepared for the simultaneous electroanalysis of IAA and SA.

## 2. Experimental

### 2.1. Reagents and Apparatus

Chitosan (CS), CH_3_COOH, C_2_H_5_OH, CuSO_4_, NaH_2_PO_4_·2H_2_O, Na_2_HPO_4_·12H_2_O, HCl, NaOH, and KCl were purchased from Sinopharm Chemical Reagent Co., Ltd., Shanghai, China, Multiwalled carbon nanotubes (MWCNTs) were purchased from Macleans Biochemical Technology Co., Ltd. Indole-3-acetic acid (IAA) and salicylic acid (SA) were purchased from Aladdin (Shanghai, China). All chemicals are analytical grade and can be used directly without further purification. Phosphate buffer saline (PBS) of 0.1 M was prepared by mixing 0.1 M NaH_2_PO_4_ and 0.1 M Na_2_HPO_4_, and the pH was adjusted by HCl or NaOH. Deionized water was used in all experiments.

CHI660E electrochemical workstation with conventional three electrode system (Shanghai Chenhua Instrument Co., Ltd., Shanghai, China,) was used for all the electrochemical experiments. Glassy carbon electrode (GCE, diameter of 3.0 mm), platinum wire (diameter of 0.2 mm) and KCl saturated calomel electrode (SCE) were used as working electrode, counter electrode, and reference electrode, respectively. Hydrothermal reaction kettle (Beijing Kewei Yongxing Instrument Co., Ltd., Beijing, China) was used for the synthesis of CNPs. FT-IR spectra was conducted on Fourier Transform Infrared Spectrometer (Bruker Company, Ettlingen, Germany). SEM images and were collected from Zeiss sigma 300 field emission scanning electron microscope equipped (Jena, Germany). TEM images, SAED images and EDX spectrum were collected from Transmission Electron Microscope (Jeol, Tokyo, Japan).

### 2.2. Procedures

Purification of MWCNTs [25] and pretreatment of GCE [26] were according to the previous report.

Preparation of CNPs was according to previous reports [27]. Briefly, CS was dissolved in a 1% CH_3_COOH solution with vigorous stirring, and the resulting solution was placed in a reaction kettle, which was then heated in an oven at 160 °C for 11 h. After cooling the samples to room temperature, the samples were taken out and dialyzed with a 3500D dialysis bag for 24 h to obtain the CNPs dispersion.

Preparation of CuNPs-CSF-CNPs-MWCNTs composite modified electrode (Figure 1). A total of 5 mg/mL CNPs were mixed with 5 mg/mL MWCNTs by sonication to obtain CNPs-MWCNTs composite dispersion. Then 6 μL CNPs-MWCNTs composite dispersion was drip-dry on the bare GCE surface to obtain CNPs-MWCNTs/GCE. Finally, the CNPs-MWCNTs/GCEs were placed in a solution of 0.04 M H_2_SO_4_ + 0.11 M CuSO_4_ at −0.4 V for 10 s, so that CuNPs were deposited on the surface of the modified electrode, while the CNPs were immobilized by deprotonation, in the modified electrode, on the surface of the electrode A thin layer of CSF was electrodeposited to prepare CuNPs-CSF-CNPs-MWCNTs/GCE. The CNPs/GCEs were placed in a −0.4 V H_2_SO_4_ solution for 10 s to obtain CSF-CNPs/GCE. The CNPs/GCE were placed in a −0.4 V H_2_SO_4_ + CuSO_4_ solution for 10 s to obtain CuNPs-CSF-CNPs/GCE. 

Optimize experimental conditions. Simultaneous detection of IAA and SA at CuNPs-CSF-CNPs-MWCNTs/GCE using linear stripping voltammetry (LSV) in 0.1 M PBS (pH = 7.0). In order to make the detection effect better, the detection conditions were optimized, including CuSO_4_ concentration, deposition potential, deposition time, CNPs concentration, hydrothermal time, hydrothermal temperature, MWCNTs concentration, PBS pH, and preconcentration time.

Determination of IAA and SA in real leaf samples. CuNPs-CSF-CNPs-MWCNTs/GCE was used to detect IAA and SA in rape leaves and broad tea leaves with standard addition method. The leaf samples were dried, ground, and soaked in methanol for 48 h, and then centrifuged to obtain a solution containing IAA and SA for detection and analysis [28].

## 3. Results and Discussion 

### 3.1. Characterization of CuNPs-CSF-CNPs-MWCNTs Composite

SEM was used to characterize CNPs-MWCNTs/GCE (Figure 1A) and CuNPs-CSF-CNPs-MWCNTs/GCE (Figure 1B). CNPs-MWCNTs/GCE (as can be seen from Appendix A, CNPs-MWCNTs were successfully prepared) were placed in CuSO_4_ + H_2_SO_4_ solution and deposited at −0.4 V for 10 s to obtain CuNPs-CSF-CNPs-MWCNTs/GCE. Comparison of Figure 1A and Figure 1B, it can be seen that a thin film is formed on the surface of CuNPs-CSF-CNPs-MWCNTs, which was obtained due to the electrodeposition of CSF by electropolymerization of CNPs fixed chitosan residues (Figure 1). 

Figure 2 shows the TEM images of CNPs (Figure 2A), CSF-CNPs (Figure 2B), CuNPs-CSF-CNPs (Figure 2C) and CuNPs-CSF-CNPs-MWCNTs (Figure 2D). Figure 2A shows that the average particle size of CNPs synthesized by CS hydrothermal reaction is 74 nm. In the TEM images of CSF-CNPs (Figure 2B) and CuNPs-CSF-CNPs (Figure 2C), it can be seen that a thin film is formed, indicating that CNPs can form CSF through electrodeposition (Figure 1), and the CuNPs are attached to the film surface. It can also be seen from Figure 2D that some CNPs can form CSF by co-electrodeposition; at the same time, CuNPs (Figure 2E,F) are attached to the surface of the composite, indicating that the CuNPs-CSF-CNPs-MWCNTs composite is successfully prepared. The composite also proved to be successfully prepared by FI-IR (Appendix A). 

As shown in Figure 3, the composite was characterized by XRD. Figure 3 shows that CNPs, MWCNTs, CNPs-MWCNTs and CuNPs-CSF-CNPs-MWCNTs have a carbon diffraction peaks (002) plane of carbon structure at about 2θ = 26°. CuNPs and CuNPs-CSF-CNPs-MWCNTs have a characteristic peak of Cu (111) at about 2θ = 43° [29]. The peak corresponding to CuNPs-CSF-CNPs-MWCNTs at around 36° is Cu_2_O (111), because the nanoparticle characterization process was carried out in the presence of normal atmosphere, that is, in the presence of O_2_ [30]. At about 2θ = 42° The left and right peaks are characteristic peaks of MWCNTs (110). This gives the idea that the surface of the CuNPs-CSF-CNPs-MWCNTs composite contains CuNPs.

Figure 4 shows CV and EIS of six different modified electrodes in 5.0 mM [Fe(CN)_6_]^3−/4−^ + 0.5 M KCl solution. The peak current of CV (Figure 4A) of the modified electrodes are in the order of CuNPs-CSF-CNPs-MWCNTs/GCE > MWCNTs/GCE > CNPs-MWCNTs/GCE > CuNPs/GCE > GCE > CNPs/GCE, and the order of EIS (Figure 4B and Table 1) of the modified electrodes are CuNPs-CSF-CNPs-MWCNTs/GCE < MWCNTs/GCE < CNPs-MWCNTs/GCE < CuNPs/GCE < GCE < CNPs/GCE, where the peak current of CNPs/GCE is the smallest and the resistance was the largest, indicating the conductivity of CNPs is poor. With the formation of MWCNTs, CuNPs and chitosan films, the charge transfer resistance of the composites decreased (Table 1). 

### 3.2. Optimization of Experimental Conditions

In order to improve the detection performance, some experimental conditions that may affect the detection effect were selected and optimized. Herein, effects of CuSO_4_ concentration (Figure 5A), deposition potential (Figure 5B), deposition time (Figure 5C), concentration of CNPs (Figure 5D), hydrothermal time (Figure 5E), hydrothermal temperature (Figure 5F), concentration of MWCNTs (Figure 5G) and the preconcentration time (Figure 5H) were optimized at CuNPs-CSF-CNPs-MWCNTs/GCE in 0.1 M PBS (pH = 7) containing 50 μM IAA and 50 μM SA. It was reported in the literature that when pH ≥ 5, there is a composite oxidation peak for IAA oxidation. This is because when the solution pH > pKa (pKa = 4.8), IAA is oxidized, resulting in the second oxidation peak [31]. Since the sensitivity of the first oxidation peak is larger than that of the second oxidation peak (Appendix A), the first oxidation peak was selected as the research object. As shown in Figure 5A, with the increase of CuSO_4_ concentration, more CuNPs adsorb on the electrode surface, the concentration is too low, resulting in too few CuNPs on the electrode surface; the concentration is too high, resulting in too much CuNPs, and the electrode surface material may easily fall off. Therefore, 0.11 M CuSO_4_ is chosen. Figure 5B shows the optimization of deposition potential, as different deposition potential may affect the morphology and size of the composite on the electrode surface [32], which in turn affects the response of the modified electrode to IAA and SA. The result shows that the oxidation peak current of IAA and SA increases with the decrease of deposition potential, and reaches the maximum at −0.4 V. When the deposition potential decreases further, the oxidation peak current decreases, so −0.4 V was selected as the optimal deposition potential. Figure 5C shows the optimization of deposition time. With the increase of deposition time, the response of composite to IAA and SA reaches the maximum at 10 s. This is because as the deposition time increases, the thickness of the electrode surface material also increases, which in turn affects the electron transport ability of the electrode, resulting in a decrease in the electrochemical signals of IAA and SA. Figure 5D is the optimization of CNPs concentration. As the concentration of CNPs increased, the response of the composite to IAA and SA reachs a maximum at 5 mg/mL. Because the concentration of CNPs will effect the thickness of CSF. The optimal hydrothermal time of CNPs in Figure 5E was 11 h. With the increase of reaction time, the number of CNPs also increases. However, with the increase of time, some CNPs are over-carbonized due to continuous heating, resulting in the destruction of their structures. Figure 5F is the optimization of hydrothermal temperature of CNPs, this is due to the low temperature, which makes insufficient reaction and less CNPs production. High temperature leads to excessive carbonization of CNPs and structural damage, so the response to IAA and SA is poor. Figure 5G shows the optimization of the concentration of MWCNTs, and the maximum response is 5 mg/mL. This may be that the higher the concentration of MWCNTs, the larger the adsorption reaction interface, the greater the electron transfer and the greater the response. However, with the accumulation of materials, the excessive thickness of materials will affect the electron transfer rate and the stability of materials loaded on the electrode surface. Therefore, 5 mg/mL MWCNTs are selected as the optimal conditions. Figure 5H shows optimization of preconcentration time, the response is maximum at 90 s. This may be due to the adsorption of IAA and SA onto the electrode surface with the increase of preconcentration time, reaching the maximum at 90 s. As the preconcentration time continued to increase, the active sites on the electrode surface gradually decreased, which affected the adsorption of IAA and SA, resulting in a decrease in their peak current responses.

Figure 5I shows the LSV responses of different modified electrodes to 50 μM IAA and 50 μM SA, respectively. It can be seen from Figure 5I that the response of CuNPs-CSF-CNPs-MWCNTs/GCE modified electrode to IAA and SA is greater than that of CuNPs/GCE, CNPs/GCE, MWCNTs/GCE, CSF-CNPs-MWCNTs/GCE and CNPs-MWCNTs/GCE modified electrode. This exhits that CuNPs, CNPs, MWCNTs and CSF increase the electroactive surface area of the composites (Appendix A), thereby enhancing its sensing performance. Therefore, the CuNPs-CSF-CNPs-MWCNTs modified electrode is selected for subsequent experiments.

### 3.3. Kinetic Behavior of IAA and SA Detection

The kinetic behavior of IAA and SA detection at CuNPs-CSF-CNPs-MWCNTs/GCE was examined, as shown in Figure 6 and Figure 7. When the scanning rate varies from 20 mV/s to 140 mV/s, the oxidation peak potential of IAA and SA shifts positively (Figure 6A), and the peak current of IAA and SA increases with the increase of scanning rate. Figure 6B,C shows that the oxidation peak current of IAA and SA has a good linear relationship with the scanning rate. The linear regression equations are *I*_pa_ (IAA_1_) = 0.2424*ν* (mV/s) + 5.1711 (R^2^ = 0.9988), *I*_pa_ (IAA_2_) = 0.1174*ν* (mV/s) + 3.699 (R^2^ = 0.9832) and *I*_pa_ (SA) = 0.0549*ν* (mV/s) + 3.336 (R^2^ = 0.9913), indicating that IAA and SA detection are typical adsorption controlled processes on modified electrode. According to the theoretical formula of Bard, A.J. and Faulkner, L.R. [33] (2022): *I*_pa_ = n^2^F^2^νAΓ*/4RT = nFQν/4RT (R = 8.314, F = 96,480, T = 298.15, ν = 100 mV/s), Q (IAA_1_) = 1.430 × 10^−5^ C, Q (IAA_2_) = 5.7 × 10^−6^ C, Q (SA) = 3.796 × 10^−6^ C, *I*_pa_ (IAA_1_) = 29.36 μA, *I*_pa_ (IAA_2_) = 16.09 μA, *I*_pa_ (SA) = 8.85 μA, The transfer electron number n for IAA_1_ and SA are both about 2. The transfer electron number n for IAA_2_ is about 3. 

Figure 7A shows the LSV curves of CuNPs-CSF-CNPs-MWCNTs/GCE at different pH values in PBS solution containing 50 μM IAA and 50 μM SA. It can be seen from Figure 7 that with the increase of pH, the peak current of IAA and SA first increases and then decreases. When pH = 7, the maximum response current is obtained. As can be seen from Figure 7A, with the increase of pH, the peak current of the two phytohormones show a negative shift, indicating the peak position of IAA and SA are closely related to pH, which may be related to the transfer of H^+^ in the solution. Figure 7B,C shows that the oxidation peak potential (*E*_pa_) of IAA_1_, IAA_2_ and SA decrease linearly with the increase of solution pH. The linear regression equations are *E*_pa_(V) = −0.0411pH + 0.9004 (R^2^ = 0.9967), *E*_pa_(V) = −0.0462pH + 1.0435 (R^2^ = 0.9910) and *E*_pa_(V) = −0.052pH + 1.255 (R^2^ = 0.9929), indicating that the redox process of IAA_1_, IAA_2_ and SA are accompanied with proton migration. According to Laviron’s (1974) [34] theoretical formula, dEp/dpH = −2.303mRT/nF, where R = 8.314, F = 96,480, T = 298.15, According to the formula in pH, m is the number of protons involved in the electrochemical reaction, and the m/n of IAA_1_, IAA_2_ and SA is 0.695, 0.781 and 0.879, respectively. The m values of IAA_1_, IAA_2_ and SA are calculated to be 1, 2, 2, respectively. The above results indicate that the electrochemical oxidation of IAA_1_ involves two-electron and a proton processes, while the electrochemical oxidation of IAA_2_ involves three-electron and two-proton processes. The electrochemical oxidation process of SA is a two-electron and two-proton process. (Figure 8), which was consistent with previous reports [1,35,36,37].

### 3.4. Detection of IAA and SA

Appendix A shows the individual detection results of IAA (Appendix A) and SA (Appendix A) at CuNPs-CSF-CNPs-MWCNTs/GCE under the optimal experimental conditions. Appendix A shows the LSV responses of different concentrations of modified electrodes to IAA. Appendix A shows the linear relationship between the peak current and the concentrations of IAA_1_ and IAA_2_. A good linear relationship is found for IAA_1_ and IAA_2_ from 0.01 to 60 μM. The linear regression equations are *I*_p_(μA) = 1.1455*c* (μmol/L) + 1.4115 (R^2^ = 0.9968) and *I*_p_ (μA) = 0.5618*c* (μmol/L) + 0.9798 (R^2^ = 0.9901), respectively, and the detection limit was 0.0078 μM and 0.0091 (*S/N* = 3), respectively. Appendix A shows the LSV responses of the modified electrodes at different concentrations to SA. Appendix A is the linear relationship between peak current and SA concentration. In the 2–45 μM range, there is a good linear relationship between peak current and SA concentration: *I*_p_(μA) = 2.012*c* (μmol/L) −2.476, (R^2^ = 0.9978) and *I*_p_(μA) = 0.9450*c* (μmol/L) +11.3310, (R^2^ = 0.9986). The detection limit was 0.24 μM (*S/N* = 3). Figure 9A shows the LSV responses of modified electrodes with different concentrations (fixed 20 μM SA) to IAA. Figure 9B shows the linear relationship between peak current and IAA concentration. A good linear relationship was found for IAA from 0.01 to 60 μM. The linear regression equation was *I*_p_(μA) = 1.1324*c*(μmol/L) + 1.0042 (R^2^ = 0.9960), and the detection limit was 0.0079 μM (*S/N* = 3). Figure 9C shows the LSV responses of modified electrodes with different concentrations (fixed 20 μM IAA) to SA. Figure 9D is a linear relationship between peak current and SA concentration. There is a good linear relationship between peak current and SA concentration in the range of 2–35 μM (*I*_p_(μA) = 1.0431*c* (μmol/L) −0.1571, R^2^ = 0.9986). The detection limit is 0.46 μM (*S/N* = 3).

Figure 10 shows the simultaneous detection of IAA and SA at CuNPs-CSF-CNPs-MWCNTs/GCE under the optimal experimental conditions. Figure 10A shows the LSV responses of modified electrodes to IAA and SA at different concentrations. Figure 10B shows the linear relationship of peak current to IAA and SA concentrations. The linear relationship of IAA in the range of 0.01–50 μM is *I*_p_(μA) = 1.0450*c*(μmol/L) + 0.8980 (R^2^ = 0.9957), and the detection limit (*S/N* = 3) was 0.0086 μM. The linear relationship of SA in the range of 4–30 μM was *I*_p_(μA) = 0.2366*c*(μmol/L) − 0.2492 (R^2^ = 0.9959), and the detection limit was 0.7 μM (*S/N* = 3). As shown in Table 2. The detection limit is superior to the simultaneous detection of IAA and SA at typical modified electrodes.

In order to investigate the anti-interference ability of CuNPs-CSF-CNPs-MWCNTs/GCE for IAA and SA detection, 0.1 M PBS (pH = 7.0) containing 50 μM IAA and 50 μM SA was added, and small molecular substances (ascorbic acid, cysteine, citric acid, arginine, glucose) and inorganic ions (Zn^2+^, K^+^) that may interfere with the experiment were added. As shown in Figure 11, there was no obvious interference compared with the peak current of IAA and SA, indicating that CuNPs-CSF-CNPs-MWCNTs/GCE has good selectivity. Figure 11 also shows the reproducibility (Figure 11B) and stability (Figure 11C) of CuNPs-CSF-CNPs-MWCNTs/GCE for IAA and SA detection. In Figure 11B, five polished electrodes were modified with the same composite and the corresponding currents were recorded. The results show that the modified electrode has good reproducibility. In Figure 11C, the modified electrodes were placed in a 4 °C refrigerator for 10 days and tested in the same solution. The results show that the current responses of the modified electrode to IAA and SA remain around 87.92% and 91.90%, respectively, indicating that the modified electrode has good long-term stability.

In order to further understand the practical value of the sensor, IAA and SA in rape leaves and tea leaves were detected by standard addition method with CuNPs-CSF-CNPs-MWCNTs/GCE. According to Table 3, the recovery rate is stable at 91.1–109%, and the RSD is 1.27–2.98%, indicating that the sensor can be applied to the detection of actual samples. 

## 4. Conclusions

In this work, a novel CuNPs-CSF-CNPs-MWCNTs composite was prepared by one-step co-electrodeposition method. CuNPs-CSF-CNPs-MWCNTs can significantly improve the conductivity and electroactive surface area of the composites explained by CV and EIS, thereby improving the performance of the sensor. The sensor is used for simultaneous detection of IAA and SA with a wide linear range and low LOD. It also has an ideal recovery rate in the detection of actual samples, so it has potential application value in the detection of IAA and SA. In addition, the proposed co-electrodeposition method can be extended to facilitate the preparation of many other composites using other CNPs fixed residue as a monomer for wide applications.

## Data Availability

The study did not report any data.

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
