# Peer review of "One-Step Co-Electrodeposition of Copper Nanoparticles-Chitosan Film-Carbon Nanoparticles-Multiwalled Carbon Nanotubes Composite for Electroanalysis of Indole-3-Acetic Acid and Salicylic Acid"

_sensors, 2022, doi:10.3390/s22124476_

Round 1
Reviewer 1 Report
Although the authors have introduced some changes in the revised document, some questions have not been satisfactorily addressed. I failed to see any innovation, with regards to synthesis design and device performance. Still, I don't recommend publishing this manuscript due to lack of clarity and in-depth investigation of the oxidation process of each hormone. Also, the merit of many components composite is not satisfactory since the improvement in signal is tiny with the addition of each component.
Yet, the presence of “Chitosan Film” in the title in not at all appropriate since the reformation of chitosan is not proved. It is only speculative that a blurred image proves the formation (reconstruction) of a chitosan film during electrodeposition. Another technique should be used to prove the structure of the reformed chitosan film simultaneous to CuNp deposition (not even the XRD experiment does not prove the reformation of chitosan).
Some redundant info in the introduction part. “IAA and SA content in plants is very low[2], sensitive to external conditions and plants contain a variety of plant hormones and plant metabolites, resulting in complex background and difficult to detect. Therefore, an efficient and specific method must be selected for detection. Therefore, it is necessary to establish a method for the determination of IAA and SA with high selectivity and sensitivity” If the lowest accepted threshold is not known, nor the upper level which become to be harmful for the plant, how can someone evaluate the normal level of these phytohormones in a plant? What is considered as control?
The authors did not discuss at all how the wrong amount can be mitigated and what measures can be applied to regulates some abnormal amounts of phytohormones. This info would support the importance of phytohormone detection.
From the figure R1 is clearly noticeable that the heights of the peaks for MWCNTs/GCE are bigger than for CNPs-MWCNT/GCE, which is also correlated to the slope of the linear equations. So, please explain why this is not reflected in the calculated electroactive surface area values.
Statement like “Along with the electrodeposition, in CuNPs-CSF-CNPs-MWCNTs, -NH2 vibration occurred in the range of 650~900 cm-1, indicating that the formation of CSF is related to -NH2” should be reconsidered since there is no relation cause-effect. And conclusion is not demonstrated by the FTIR experiment.
Figure 2C is hardly understandable “HRTEM image of the CuNPs-CSF-CNPs-MWCNTs composite, which further proves the existence of CuNPs with a lattice of 0.23 nm” since the CuNp have a distinguishable shape and sizes about tens of nm. Still not explained the presence of “lattice”.
Statement like ”Since the first oxidation peak is more sensitive than the second one, the first oxidation peak was selected for electrochemical detection of IAA” should be reformulated and the entire phenomenon should be reconsidered. It is not about the sensitivity of any peak, it is about the electrochemical oxidation mechanism of indoles, whish occurs in two-steps and are not clearly attributed (not in this section). The oxidation of IAA is very careless treated since it is a complex process that should be investigated first individually, then together with SA.
Trivial expressions like “It can be seen from Figure 1D and Figure 2B that CuNPs-CSF-CNPs-MWCNTs composites contain CuNPs, CNPs, MWCNTs and CS” should be avoided.
All the optimizations put together in figure 5 are just observations of obtained results, but not explained why they happen and what is the reason. Also, in caption of figure 5, there is a talk about an Insert figure, which does not exist.
Fig. 10 – the peak oxidation of SA is difficult to differentiate and measure in the presence of IAA, which makes the applicability of the sensors doubtful. They should be investigated separated first, then in a mixture.
Reviewer 2 Report
The authors have given quite an effort to improvise the previous version of the manuscript by responding to all the comments and it can be now published in its revised form. I believe it will grab the attention of the readers if published.
Reviewer 3 Report
Dear Authors,
please find the comment enclosed below.
Sincerely,
The Reviewer.
Lines 40-42
Rephrase trying to be more clear.
“Therefore, an efficient and specific method must be selected for detection. Therefore, it is necessary to establish a method for the determination of IAA and SA with high selectivity and sensitivity.”
Lines 44-45
Check the language style
“Among them, electrochemical sensors have the advantages of simple, sensitive, rapid and economical, which are the most promising sensors”
Line 46
A space is missing, even in other parts along the manuscript (e.g. Line 63)
“Chitosan (CS) [10]is…”.
Lines 62-64
Very hard to understand what Authors mean. Please rephrase
Lines 93-99
Being the electrodeposition method common to all the substrate investigated, this part could be rearranged in a clearer perspective avoiding to be repetitive.
Lines 101-103
“The detection conditions were optimized…”
According to that, why include again in this part the method optimization for the sensor preparation? This could induce confusion in the reader.
Please, rephrase according to the introduction of this period.
Lines 130-137
Deep revision of the language style is needed
Line 132
“The MWCNTs/GCE ((Figure 1B) had a smoother surface than the CNPs-MWCNTs/GCE ((Figure. 1C)…”
The SEM images shown are very similar among them, at least at this level of magnification, don’t allowing to support what authors state. EDX/EDS analysis would be helpful.
Furthermore, what scale is nM, nanoMolar? Please check and correct it with nanometers, nm.
Lines 136-137
Following the previous comment, the quality of the overall Figure 1 is very low, probably due to the combining of 4 SEM images in a single small Figure. To overcome this limit, please consider to split these SEM images, leaving the most significative in the main document and transferring the others for the comparison in a supporting information/supplementary data document.
Furthermore, even in this case (Fig. 1-D), according to the SEM characterization shown, the presence of the CuNPs is barely distinguishable. Especially considering the comparison with figure 1-A, where the size of the NPs achieved in the same electrodeposition conditions are quite different, ~500 nm apparent diameters in Fig. 1A Vs around ~100 nm in Fig 1D. Please, discuss this evidence.
The presence of the CuNPs can be addressed only going forward along the manuscript to Figure 2C where TEM characterization is shown. Referring to this latter is available the SAED image of the CuNPs to support their composition?
Lines 166-170
Following the previous comment, the SAED image of the CuNPs in fig 2C would be helpful considering XRD data shown (Fig.3-A). By the comparison among the spectra of the bare CuNPs (magenta line) and the composite CuNPs-CSF-CNPs-MWCNTs (red line), two further peaks in the 2theta theta region among 0 and 43 degrees and among 35 and 40 degrees are distinguishable. This was not addressed, please discuss it including the comparison with the following literature reference:
M.P. Sooraj, Archana S. Nair, Suresh C. Pillai, Steven J. Hinder, Beena Mathew, CuNPs decorated molecular imprinted polymer on MWCNT for the electrochemical detection of l-DOPA, Arabian Journal of Chemistry, Volume 13, Issue 1, 2020, Pages 2483-2495
Moreover, as a general suggestion, representing XRD data together with crystallographic database reference patterns would help to easily address all the peaks found.
Lines 205-214
The equivalent circuit model shown in Fig. 4-B, from which are derived data shown in Table 1, is characterized by the presence of a Capacitor (C). Why did you use this element instead of the constant phase element (Q or CPE)? Please discuss.
Furthermore, in Table 1 is shown only the estimated charge transfer resistance while the other parameters derived by the equivalent circuit are missing. The same for the chi-square indicating the goodness of the non linear least square fit performed. Please add them in the same table.
Referring again to the EIS characterizations of the same figure, why did you perform them at 0.2 V? Please discuss
Lines 212-213
“As shown in Table 1, the results showed that the formation of chitosan film could improve 212 the electron transport capacity of the composites”
To avoid confusion in the reader, in the opinion of this Reviewer, is more accurate talking of “decreased charge transfer resistance” for the CuNPs-CSF-CNPs-MWCNTs/GCE, as in Lines 210-211.
According to that lines 212-213 are unnecessary.
Lines 225-227
“because different 225 deposition potential affects the morphology and size of composite on the electrode surface, and then affects the response 226 of the modified electrode to IAA and SA.”
As a suggestion, if available consider the possibility to add SEM characterization emphasizing it in a supporting information/supplementary data document.
Line 303
Please correct the name of Allen J. BARD and add the citation for that book (Bard, A. J., Faulkner, L. R., & White, H. S. (2022). Electrochemical methods: fundamentals and applications. John Wiley & Sons.).
Line 340
Add the reference.
Reviewer 4 Report
- The authors should clearly inform the readers what each of these CuNPs-CSF-CNPs-MWCNT is? For example reading " CNPs were prepared by hydrothermal method using CS as carbon source" am trying to understand what CNP is?
- How does the morphology of the CuNPs or the synthesized material contribute to the sensing application?
- The authors mention that the first oxidation peak is more sensitive than the second one, the first oxidation 222
peak was selected for electrochemical detection of IAA. Why is the first oxidation peak sensitive? - How was the surface area of these samples calculated?
- Make some grammatic and spacing errors.
Round 2
Reviewer 1 Report
The paper is much improved in this format. I recomand the acceptance for publication.
This manuscript is a resubmission of an earlier submission. The following is a list of the peer review reports and author responses from that submission.
Round 1
Reviewer 1 Report
The authors reported a CuNPs-CSF-CNPs-MWCNTs composite used for electroanalysis of Indole-3-Acetic Acid and Salicylic Acid. The paper abounds of confusing statements and lack in scientific novelty and analytical progress.
I do not recommend publication of this work in Sensors journal and my comments for improving the quality of this work are shown below:
- Too scarce introduction. Authors state about the phytohormone „If they are wrong in the amount of substances at a certain stage, they will lead to plant diseases” but not mention what is the purpose/importance to determine both phytohormones in plants? And what kind of measure can human apply to regulates some abnormal values? Or how the wrong amount can be mitigated?
- What is the reason to use two carbon materials? It is suppose that carbon materials increases the active surface area and conductivity of the electrode, so they should explain and motivate the usage of two material from the same class.
- trivial expression when explaining the experiments
- very unclear the steps of electrode modification „CNPs/GCE was prepared by drip-dry 6 mL composite dispersion with MWCNTs”. Also for „ MWCNTs/GCE was prepared by drip-dry 6 mL composite dispersion with CNPs”. Which dispersion for each modification?
- confusing expression „CuNPs/GCE was prepared by electrodeposition of GCE for 10 s..” . Metals will accomplish the electrodeposition process through reduction, not the GCE. The electrode is just the surface onto which the metallic nanoparticles will be deposited allowing the electronic transfer. All experimental part should be reformulated.
- it is not clear where is chitosan involved in the preparation of modified electrode?
- Fig 1A – what is the substrate onto which CuNp are deposited? they should be shown as deposited onto CNPs-MWCNTs since the substrate has a huge influence on the nucleation and growth of metallic nanoparticles
-what was actually the conclusion drawn from figure 1? How they prove the polymerization really happen? Why the CuNp are not visible in fig 1D if they have sizes of 400-500 nm as seen in fig 1A?
- „ the lattice of CuNPs is 0.3195 nm (Figure 2C)” this statement should be reconsidered and explained. Did the CuNp form a lattice since they are shown as distinct nanoparticles of bigger size in fig 1A?
-Unclear CV experiment. The influence of each component of the composite should be demonstrated toward detection of phytohormone
- detection of IAA and SA should be treated separately first to establish conditions for each of them, then as simultaneous detection
- there is an oxidation peak around 0.7 V in all LSV experiments which is not even mentioned and properly attributed
- Why would someone choose such a many-components composite for preparation of a modified electrode? However, the article is difficult to read, with repetitive expressions and not well structured presentation of the protocol.
All these points need attention in order to clarify some of the argumentation.
Reviewer 2 Report
Comments to the Authors: Manuscript ID, sensors-1645869
The manuscript submitted by the author Kuang et al. on “One-Step Electrodeposition of Copper Nanoparticles-Chitosan Film-Carbon Nanoparticles-Multiwalled Carbon Nanotubes Composite for Electroanalysis of Indole-3-Acetic Acid and Salicylic Acid”, developed for detecting selectively phytohormones indole-3-acetic acid (IAA) and salicylic acid (SA) is quite good and will improve the present state of the art. The authors have performed a thorough study following a detailed analysis. Therefore, the work done is comprehensive as research development and has opportunities in future applications.
I believe the work is quite interesting and can be published in Sensors with minor corrections.
- The Cu nanoparticles shown in Figure 1A are not properly visible. The figure should be given with a higher resolution.
- Figure 2 should be with higher clarity. The images are not very clear.
Reviewer 3 Report
The author reports a sensitive electroanalysis procedure of the phytohormones indole-3-acetic acid (IAA) and salicylic acid (SA) based on a novel Cu nanoparticle-chitosan film-carbon nanoparticle-multiwalled carbon nanotube composite (CuNPs-CSF-CNPs-MWCNTs). The CNPs were prepared by the hydrothermal method followed by the preparation of CuNPs-CSF-CNPs-MWCNT by electrodeposition of CNP-MWCNT electrode in CuSO4 solution. The composite electrode analyzed the IAA in the range 0.01-50 μM, and SA in the range 4-30 μM. The detection limits were 0.0086 μM and 0.7 μM for IAA and SA, respectively. The sensor can be used for the simultaneous detection of IAA and SA in real leaf samples with sufficient recovery. However, some revisions are necessary before this paper can be accepted as detailed below:
- Some Grammatical and typo errors need to be checked and corrected
- Can the author mention the intermediate anion produced as reported in Fig. 5, line 210?
- SEM and TEM image quality should be improved
- In line 164, the representation of h k l planes should be corrected.
- In lines 317, 345, and 384 specific figure numbers should be included.
- Why is chitosan more significantly used for biomedical applications when compared to other biopolymers?
- Some recent literature can be cited. (10.1007/s43207-020-00088-z; 10.3390/app9010016; 10.3390/coatings11121563)